# Design and Integration of the Single-Lens Curved Multi-Focusing Compound Eye Camera

**DOI:** 10.3390/mi12030331

**Published:** 2021-03-21

**Authors:** Kekai Tao, Gaoge Lian, Yongshun Liu, Huaming Xing, Yi Xing, Xiangdong Su, Xin Feng, Yihui Wu

**Affiliations:** 1State Key Laboratory of Applied Optics, Changchun Institute of Optics, Fine Mechanics and Physics (CIOMP), Chinese Academy of Sciences, Changchun 130033, China; taokekai18@mails.ucas.ac.cn (K.T.); lian_gaoge@163.com (G.L.); xhm3653@163.com (H.X.); xinyi18@mails.ucas.ac.cn (Y.X.); fengxin20@mails.ucas.ac.cn (X.F.); 2University of Chinese Academy of Sciences, Beijing 100039, China; 3Shenyang Ligong University, Shenyang 110159, China; directcg@msn.com

**Keywords:** curved compound eye, ommatidium, multiple focal lengths, high resolution

## Abstract

Compared with a traditional optical system, the single-lens curved compound eye imaging system has superior optical performance, such as a large field of view (FOV), small size, and high portability. However, defocus and low resolution hinder the further development of single-lens curved compound eye imaging systems. In this study, the design of a nonuniform curved compound eye with multiple focal lengths was used to solve the defocus problem. A two-step gas-assisted process, which was combined with photolithography, soft photolithography, and ultraviolet curing, was proposed for fabricating the ommatidia with a large numerical aperture precisely. Ommatidia with high resolution were fabricated and arranged in five rings. Based on the imaging experimental results, it was demonstrated that the high-resolution and small-volume single-lens curved compound eye imaging system has significant advantages in large-field imaging and rapid recognition.

## 1. Introduction

At present, optical systems are developing in the direction of compact structures, integration, and large fields of view. Natural compound eyes have the characteristics of a large field of view, small size, and high sensitivity to moving objects [1]. Inspired by natural compound eyes, bionic artificial compound eyes have become a focused area of research in recent years. Researchers have developed a variety of compound eye processing technologies, such as ultra-precision processing [2,3], laser direct writing [4,5], femtosecond laser etching [6], molding processes [7,8], and bottom-up technology [9,10]. However, the development of single-lens curved compound eye imaging systems has just started, and there are still some technical obstacles to be resolved, including the defocus between the lens and the image sensor and the size and resolution of the ommatidia of the artificial compound eye.

Natural compound eyes exist in insects such as dragonflies and fruit flies, but they have nonplanar retinae that match them [11,12]. Therefore, they do not have the phenomenon of defocus, and artificial compound eyes also require photosensitive components that match the focal plane. However, commercial image sensors are mostly flat image sensors, and the manufacturing process of curved sensors is complicated and costly [13,14]. Therefore, it needs to be solved by other methods. At present, there are generally two ways to solve the above problem. One is adding a relay system between the compound eye and Complementary Metal-Oxide-Semiconductor(CMOS), which is similar to the stalk in a natural compound eye, such as freeform prisms [15,16], optical fibers [17,18], and a lens group [19,20]. Another method is to adopt the design of the nonuniform curved compound eye, where the focal length of the ommatidium of each ring of the curved compound eye is different, and they fall on the same plane [21]. 

Many methods have been proposed for preparing nonuniform curved compound eyes, such as thermal reflow [22] and inkjet droplets [23]. Due to concerns about the volume loss during the photoresist melting process and the consistency of microdroplets in inkjet droplets, our laboratory has developed a contactless polymer hot embossing method [24]. However, in this method, the contour of the ommatidium is formed entirely by surface tension of the polymer at the glass transition temperature. Once the surface tension limit is exceeded, the large sag height is no longer achieved. According to the calculation, this will greatly limit the resolution of the curved compound eye. 

Based on the previous research of our laboratory [25], a two-step gas-assisted process is proposed in this article. It is combined with photolithography, soft photolithography, and UV curing. In the first step, the polydimethylsiloxane (PDMS) film is covered on the microhole array to prepare a planar microlens array. In the second step, the planar microlens array is transformed into a curved compound eye. It uses the relationship between the deformation of the film and the air pressure, and an ommatidium with a large height and large diameter can be achieved by changing the diameter of the ommatidium and the value of air pressure. The lens fabricated by this method shows great uniformity and high resolution. In order to promote further research into the practical application of the curved compound eye, a single-lens curved compound eye camera is integrated, as shown in Figure 1a. This camera comprises the following parts: Lens holder, lens, lens holder base, side walls, and CMOS, as shown in Figure 1b. The lens is fixed in the lens holder and the holder is connected with the base by thread. All the remaining parts are assembled by screws. In the imaging experiment, the curved compound eye imaging system can realize the large field-of-view imaging and recognize objects rapidly, which shows that it can be applied in many fields.

## 2. Lens Design and Fabrication

It was found that for the ommatidia of different diameters, under the same pressure, the sag height obtained by the gas-assisted process is approximately proportional to the diameter of the ommatidium. For example, the simulation under 8000 Pa is shown in Figure 2.

Figure 3 shows the design of the ommatidium. As all the ommatidia are arranged in a circle, each ommatidium on the same ring is symmetrical around the center and has the same focal length, so it is sufficient to select one ommatidium for the analysis on each ring.

The ommatidia are arranged from the center to the periphery; the central ommatidium corresponds to the first ring, and the outermost ommatidia correspond to the nth ring. For the nth ring, the chord length of the ommatidia will be $s_{n}$, and the sag height will be $h_{n}$, while the radius of curvature of the ommatidium $r_{n}$ can be expressed by Equation (1):(1)$$r_{n}=\frac{s_{n}{}^{2}+h_{n}{}^{2}}{2h_{n}},$$

If the pressure remains unchanged, $h_{n}$ can be calculated according to the linear relationship after determining the value of $s_{n}$. Substituting the value of $r_{n}$ from Equation (1) into Equation (2), the focal length of the nth ring can be expressed by Equation (2):(2)$$l_{n}=f^{\prime }=\frac{r_{n}}{1-m},$$
where m is the refractive index of the lens material and θ is the angle between the optical axis of the first ring and the optical axis of the Nth ring. R is the radius of curvature of the plano-convex base and can be expressed as:(3)$$\theta =\arccos\frac{R+l_{1}+h_{1}}{R+l_{n}+h_{n}},$$
representing the position of the nth ring on the plano-convex lens.

The resolution of sub-eyes can be determined according to the radius of the Airy disk, and it can be expressed as:(4)$$r_{A}=1.22\frac{\lambda }{d/f},$$

If the wavelength is a fixed value, the larger the value of d/f, the higher the resolution. Substituting Formulas (1) and (2) into Formula (5), d/f can be expressed as Equation (5):(5)$$\frac{d}{f}=(1-m)\frac{8h_{n}d}{d^{2}+4h_{n}{}^{2}},$$
where (1 − m) is a fixed value and d is the value of the diameter of the ommatidia. Equation (5) is shown in Figure 4.

It can be seen from Figure 3 that the larger the sag height, the smaller the diameter, and the higher the resolution. However, increasing h and decreasing d will shorten the focal length. There are glass packages on the surface of commercial CMOS detectors, which will make the focal plane of the lens unable to reach the surface of the detector. Additionally, according to the design criteria, if the diameter is too large, the number of ommatidia that can be arranged is lower, which reduces the acuity and filling factor to a certain extent. Therefore, five values in the range of 300–400 μm are selected for principal verification. Their diameters are 310, 312.8, 321.8, 336.2, and 358.8 μm, and the sag heights are, respectively, 42, 42.56, 44.36, 47.24, and 47.76 μm.

The process of fabricating the curved compound eye is mainly divided into two parts. First, the planar microlens array (MLA) is fabricated. Preparing the photomask with the designed pattern we need, and using photolithography, we can transfer the pattern onto the silicon wafer. The photoresist chosen in the experiment is one of the positive photoresists—AZ5214. It is spin-coated onto the silicon wafer at a speed of 2000 r/min for 45 s. Then, in the step of soft baking, we put the spin-coated silicon wafer onto a hot plate, which is heated up to 120 °C, for 120 s. After that, a mask alignment system (SUSS MA6/BA6) is used to expose the photoresist for 40 s, with the prepared mask above the wafer. The silicon wafer is immersed in the developer to remove the exposed part of the photoresist until the expected MLA structure is obtained. Before the process of hard baking, we need to clean the silicon wafer with deionized water to remove residual developer and impurities. The temperature and time used in hard baking are the same as those used in soft baking.

Next, the silicon wafer needs to be etched by plasma in the plasma etching machine (Alcatel 601E). The pattern of the photomask has already been transferred to the photoresist; thus, the patterned photoresist will serve as the mask in the etching process, transferring the pattern onto the silicon wafer. The thickness of the silicon wafer is 380 microns, and it needs to be fully etched—that is, the etching height should be 380 microns. After the etching is completed, the silicon wafer is cleaned with acetone to remove unexposed photoresist and organic substances used in the etching process. It is put in the boiling piranha solution (the ratio of sulfuric acid and hydrogen peroxide is 3:1) and then rinsed with deionized water, repeating the above process three times to complete the final cleaning.

Figure 5 shows the process of fabricating the planar MLA. The silicon wafer with the planar microhole array is placed on the vacuum cavity mold as in Figure 5a, and it is covered with a layer of PDMS film, as shown in Figure 5b. The mold possesses two tubes—one is connected to the barometer, and the other channel is connected to the suction pump. The piston is pulled, and under the action of negative air pressure, the PDMS film will deform. When the pressure is controlled precisely, we can obtain the expected shape. The curing adhesive (NOA 63) is dropped on the PDMS film to replicate planar MLA of the PDMS film, as shown in Figure 5c, and the planar MLA is achieved.

After obtaining the planar MLA, it is fixed in the prepared PMMA mold, as shown in Figure 6. The PDMS droplets are dropped into the mold and placed in a 60 °C environment for thermal curing for 12 h. After the PDMS solution curing, a thin replicated film with a reversed MLA of the planar MLA can be achieved.

Figure 7 shows the manufacturing process for the curved compound eye. The PDMS film with a concave MLA is placed on another vacuum mold. As shown in Figure 7a, the mold also possesses two tubes. There is a circular step with a diameter of 30 mm and a height of 1 mm on the upper surface to position the PDMS film so that the PDMS film is located in the center of the mold. The desired radius of curvature can be obtained by controlling the pressure inside the cavity. Under negative pressure, the PDMS film will deform into a spherical profile. Then, the UV curing adhesive NOA63 is poured into the concave cavity, and a piece of quartz glass with a diameter of 25.4 mm is used to cover the top. After irradiating under the ultraviolet lamp for 5 min, we can obtain the curved compound eye we want.

In this process, there are several points to note: The pressure value of the vacuum chamber directly affects the contour of the curved compound eye. Therefore, it is necessary to accurately control the pump suction according to the value of the barometer. The tightness of the mold is very important, and silicone grease is used to ensure its airtightness. At the same time, when the UV curing adhesive is dropped, if the amount of the UV curing adhesive is too great, it will overflow when covering the quartz glass, which will affect the surface finish of the quartz. If the amount of the UV curing adhesive is too small, it will also cause surface defects.

## 3. Result and Discussion

### 3.1. Shape Measurement

The shape measurement of the compound eye is mainly the measurement of diameter and height. The diameter of the ommatidium is measured with an ultra-depth-of-field microscope (KEYENCE, VHX-1000, Osaka, Japan). Figure 8b shows the results of the measurement, and the values of each ring are 153.74, 156.83, 161.33, 167.61, and 179.28 μm. The bright ring is the image of the microscope light source formed by the MLA. Comparing the actual value obtained by the measurement with the theoretical value of the design, the deviation is less than 5%. The equipment used to measure the sag height is a step meter (Veeco Dektak 150). The probe scans along the center of the ommatidia and the results of the actual height are shown in Figure 8c. The sag height of each ring is 40.55, 42.8, 44.5, 46.5, and 49 μm, respectively. These data are approximately linear, as seen in Figure 8d, and the deviation is less than 5%. In the process, there are two steps that are prone to errors. First, because PDMS is an elastic material, it shrinks after curing and demolding. The other is that when the PDMS film converts from a flat surface to a curved surface, it stretches. However, the thickness of the film is much greater than the height of the ommatidium, so these two errors will be compensated, and the total error will decrease. In previous experiments performed by our laboratory, it was found that the 1 mm-thick film can meet the fabrication requirements. At the same time, the operation of the measuring researcher and the measurement accuracy of the instrument will also produce some errors. In general, this result of the measurement shows the high accuracy of this fabrication method.

Finally, the stereoscopic microscope (OLYMPUS SZ61) is used to observe the surface of the ommatidium. As shown in Figure 8e,f, the surface is smooth and clean. We can draw a conclusion from the measurement results that the two-step gas-assisted process can fabricate the curved compound eye perfectly.

### 3.2. Optical Test and Characterization

According to the design, to correct the defocus phenomenon, the focal lengths of the ommatidia of different rings are different. In order to confirm whether the actual results are consistent with the design expectations, the planar MLA is placed on the imaging test platform for the imaging experiment, and the images are shown in Figure 9.

When adjusting the distance between the lens and CMOS, when the central ommatidium is in focus, the central image is clear, as shown in Figure 9a. As it expands to the edge, the image clarity of the peripheral ommatidia gradually decreases. Then, if the lens is kept away from the CMOS until the images of the outermost ommatidia are clear, the clarity gradually decreases from the periphery to the center. Obviously, the focal lengths are different—the closer the periphery, the longer the focal length.

In the above experiment, we have proved that each ring has a different focal length. When the planar MLA is converted into the curved compound eye, we need to further demonstrate that the focus of each ring has fallen on the same plane. There are generally two ways to complete the demonstration. One is the interferometric technique [26], and the other is the setup used in this article.

The focal length measurement generally uses a collimated laser setup to test the focus and light intensity. The setup is shown in Figure 10a. From left to right, it is composed of a laser, a small hole, a damping film, a curved lens, and a CMOS camera. The parallel light emitted by the laser passes through the curved compound eye; then, by adjusting the distance between the lens and CMOS until the focal spots are sharp, the images of the spots are received by the CMOS. Figure 10b shows the captured focal spots. When the focal spots can be captured clearly, the distance is 2 cm, and we can consider that the focal length of the compound eye is 2 cm. As shown in Figure 10c, the distribution of the intensity is uniform, which means that the focal spot of each ommatidium has fallen on the same plane. According to the image of the focal points, the point spread function (PSF) of each ommatidium can be obtained. The PSF can characterize the resolution of the ommatidium. Taking the flanking ommatidium as an example, its point spread function is shown in Figure 10d. The spot radius magnified by a 5× objective occupies four pixels, and the size of each pixel is 5.3 µm. Therefore, the radius of the spot is 4.12 µm. The resolution target (THORLABS 1951 USAF R3L3S1N) is used to measure the actual resolution of the ommatidium, and the result is shown in Figure 10e. The highest resolution of the ommatidium is 17.95 lp/mm. In the previous study, the maximum resolution of the nonuniform curved compound eye prepared by contactless polymer hot embossing could only reach 8.95 lp/mm. Compared to that, the results show that the compound eye prepared by the two-step gas-assisted method has a higher resolution, which will be beneficial for image stitching and image recognition. However, it is found that there is a deviation between the value of the PSF and the actual value of the resolution test. This is because the light source used to measure the spot is a single-wavelength collimated laser, so the aberration is relatively small. However, the light source used to measure the resolution is white light, which causes chromatic aberration. At the same time, the larger the field of view of the ommatidium and the larger the angle of incident light, the more serious the coma and astigmatism [27]. Therefore, aberrations have a great impact on imaging and need to be reduced. It is well known that aspheric lenses can reduce aberrations without increasing the volume of the imaging system [28]. In future work, the current spherical cap-shaped ommatidium can be turned into an aspheric ommatidium by increasing the air pressure. 

Another setup to measure the field of view of the compound eye is shown in Figure 10f. Some letters and symbols are printed on the shadowless lamp board, and the lamp is placed on the opposite side of the compound eye. The image captured by the camera is shown in Figure 9g, and we can see that the letters the camera can capture are from 5 to 9. The distance between the camera and the lamp board is 8 cm, the distance between letters 5 and 9 is 20 cm, and the field angle of the curved compound eye can be calculated as 86°. As this compound eye has only five rings of ommatidium, its field of view does not exceed 100°. If the number of the rings is increased, a larger field of view (FOV) can be achieved.

### 3.3. System Imaging and Processing

Take the classic Lena diagram as an example. The image captured by the curved compound eye system is shown in Figure 11a—each ommatidium has its own field of view, each ommatidium can be an independent imaging unit, and the distance between each ommatidium is large enough, which prevents the appearance of a ghost image. Every sub-image is only part of the target and the neighboring sub-images have some duplicate regions. Based on these duplicate regions, the image of the whole target can be achieved when all the sub-images are stitched, as shown in Figure 11b. Considering the high sensitivity of the compound eye to the moving object and using deep learning on the compound eye image, the artificial compound eye can be used for fast recognition in a large field of view. Deep learning is an important part of artificial intelligence and is currently widely used in computer vision. YOLO (You Only Look Once) is a new target detection method of deep learning, which is characterized by high speed and high accuracy. The image of each ommatidium is inputted into a convolutional neural network to detect the target separately. The test results are shown in Figure 11c,d. At the same time, it is found that the image of the compound eye contains rich direction information. In the pre-research, a 10 cm × 10 cm rectangular white block is used as the detection target. When the target moves, the image captured by the compound eye presents a pattern. Take the right direction of the camera image as the starting position and divide it clockwise into 360°. The target is moved 3 cm to the left and right, respectively, and the captured image and the analysis result are shown in Figure 11e,f. When the target is moved 3 cm to the left, there is a large offset in the range of 0° to 60° and 250° to 360°. When the target is moved 3 cm to the right, there is a large offset in the range of 120° to 250°. The closer the ommatidium is to the periphery, the higher the sensitivity to motion information and the greater the offset. As a multi-eye vision camera, the compound eye camera can provide greater depth information compared with a binocular vision camera. In future research, this could be used to obtain the three-dimensional information and velocity vector of the target, which are important for rapid recognition.

## 4. Conclusions

In general, a multi-focusing curved compound eye is fabricated by the two-step gas-assisted process and integrated into a single-lens curved compound eye camera. Through the nonuniform design, the defocus problem of the curved compound eye is solved. Using the deformation relationship between the air pressure and the film, the ommatidia with large diameter and large sag height are accurately prepared. A total of 61 sub-eyes are arranged in five circles on the plano-convex base, the diameters of the ommatidia range from 153.74 to 179.28 μm, the sag height ranges from 40.55 to 49 μm, and the resolution of the ommatidium can reach up to 17.95 lp/mm. The curved compound eye prepared in this paper has good uniformity and small error, and the field of view angle reaches up to 86°. The advantages of the curved compound eye in large field-of-view imaging and rapid recognition mean that it has great potential in a wide range of applications, such as the detection, military, and medical fields. However, in future research, it will still be necessary to further improve the imaging quality of the curved compound eye, optimize the algorithm, and shorten the response time.

## Figures and Tables

**Figure 1 micromachines-12-00331-f001:**
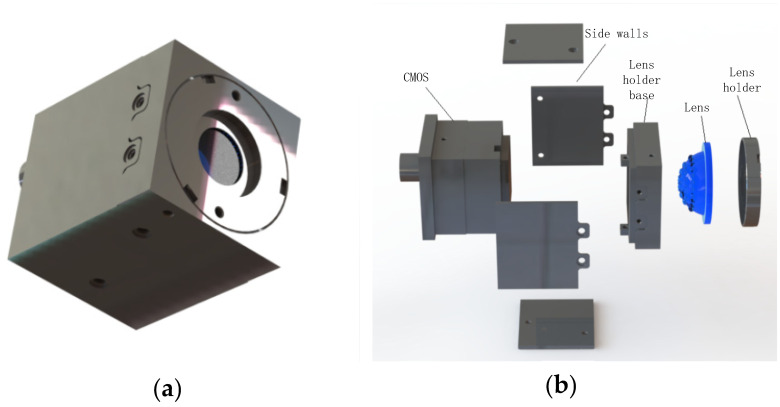
Three-dimensional massive structure of the compound eye camera. (**a**) Photo of the prototype; (**b**) exploded view of the camera.

**Figure 2 micromachines-12-00331-f002:**
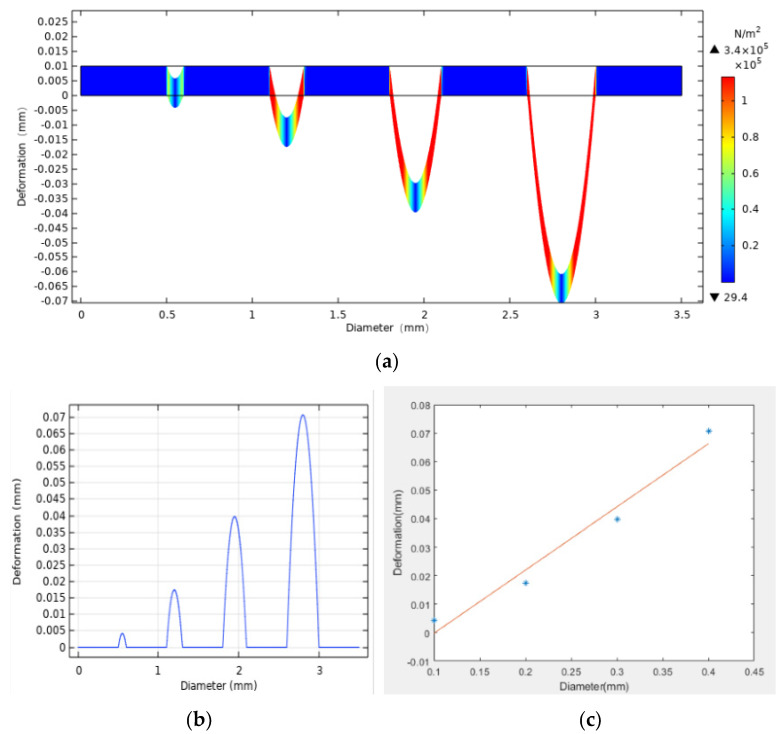
Schematic view of the results of the simulation. The thickness of the film is 10 μm, the pressure is 8000 Pa, and the diameters of the ommatidia are 0.1, 0.2, 0.3, and 0.4 mm. (**a**,**b**) The deformation of the ommatidia; (**c**) the linear relationship between the deformation and the diameters.

**Figure 3 micromachines-12-00331-f003:**
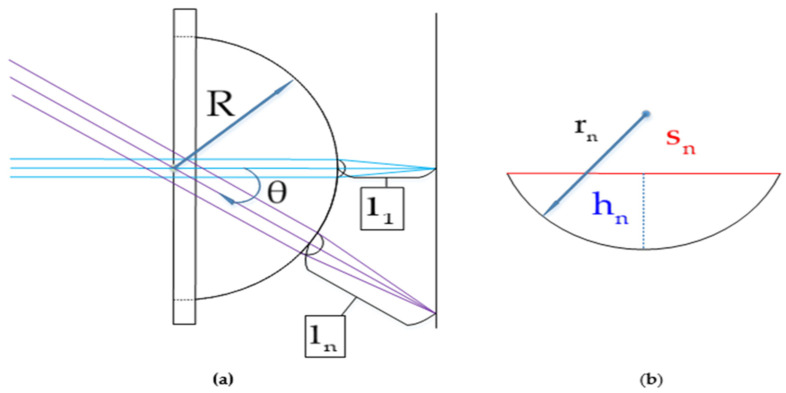
Schematic view of the multi-focusing compound eye. (**a**) Description of the design of the lens; (**b**) the parameters of the single ommatidium.

**Figure 4 micromachines-12-00331-f004:**
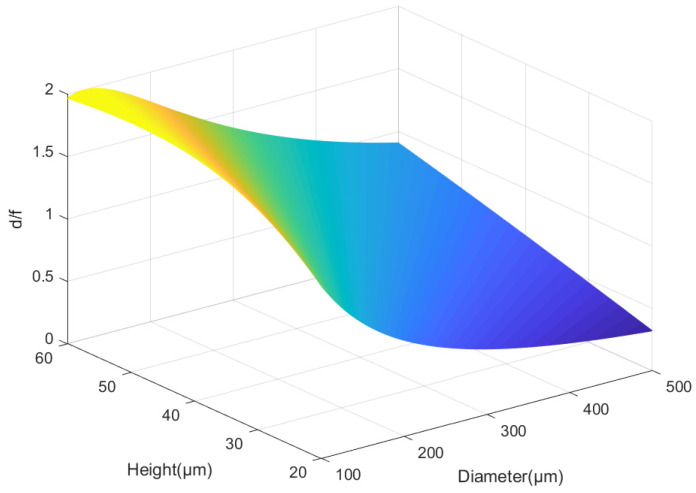
The function image of Equation (5).

**Figure 5 micromachines-12-00331-f005:**
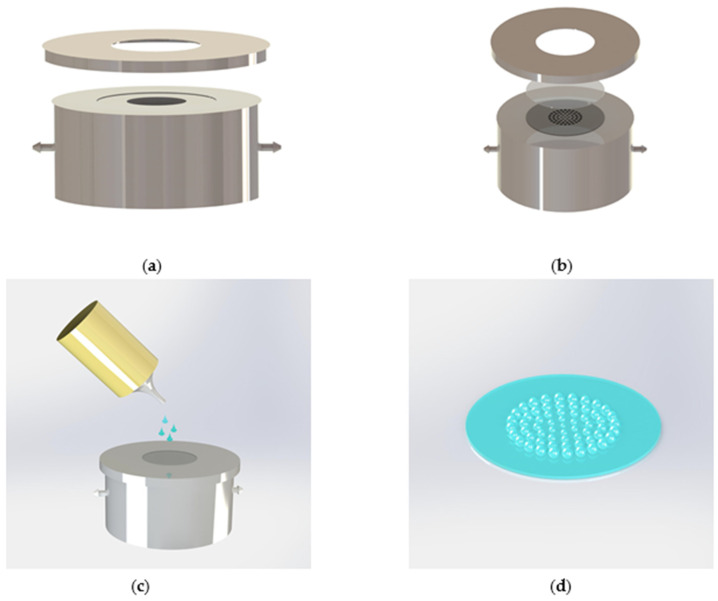
Fabrication of the planar microlens array (MLA). (**a**) The homebuilt gas-assisted mold to fabricate the planar MLA; (**b**) the PDMS film and the silicon wafer with MLAs are placed in the mold; (**c**) the UV adhesive (NOA 63) is dropped under negative pressure; (**d**) the fabricated planar MLA.

**Figure 6 micromachines-12-00331-f006:**
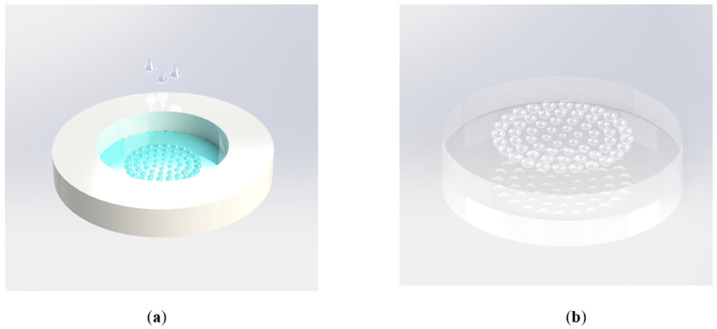
(**a**) Schematic illustration of the soft lithography process; (**b**) the fabricated PDMS film with concave MLA.

**Figure 7 micromachines-12-00331-f007:**
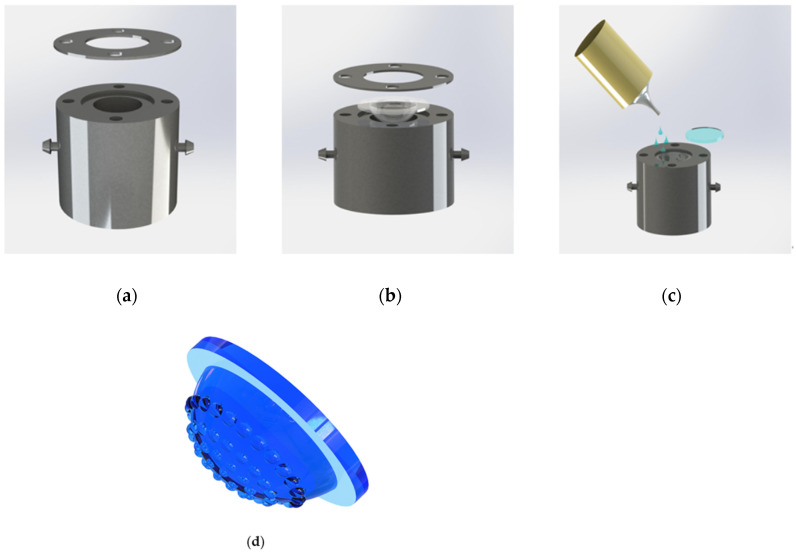
(**a**) The homebuilt gas-assisted mold to fabricate the curved compound eye; (**b**,**c**) schematic view of the process to convert the PDMS planar MLA into the curved shape; (**d**) the manufactured curved compound eye.

**Figure 8 micromachines-12-00331-f008:**
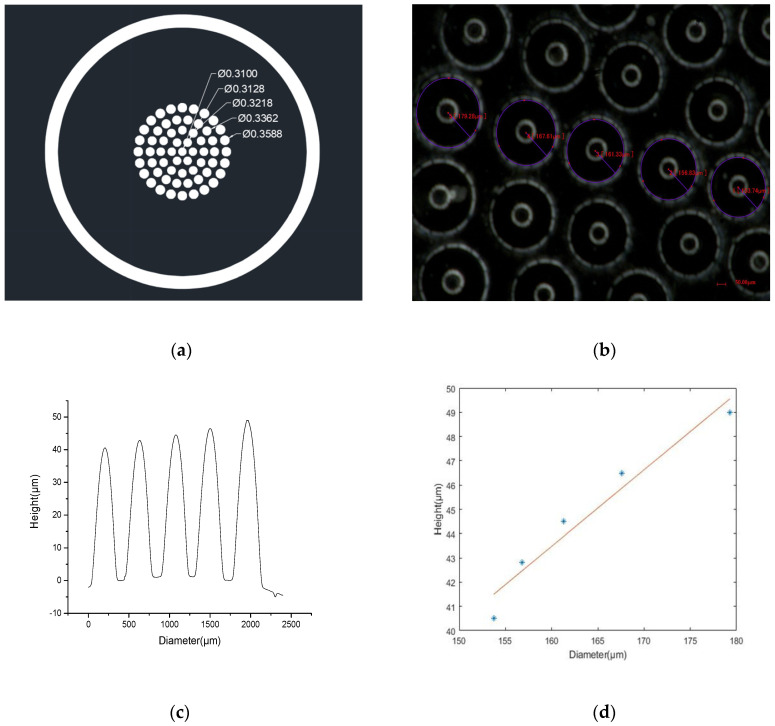
Measurement of the ommatidium. (**a**) The designed MLA on the mask; (**b**) the actual MLA; (**c**) the sag height of the ommatidium; (**d**) the linear relationship between the diameters and the height of the ommatidium; (**e**,**f**) the surface of the compound eye.

**Figure 9 micromachines-12-00331-f009:**
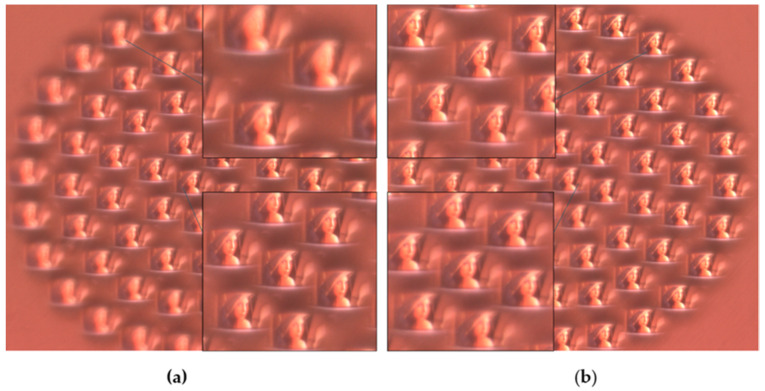
(**a**) The captured image when the central ommatidium is in focus; (**b**) the captured image when the flanking ommatidia are in focus.

**Figure 10 micromachines-12-00331-f010:**
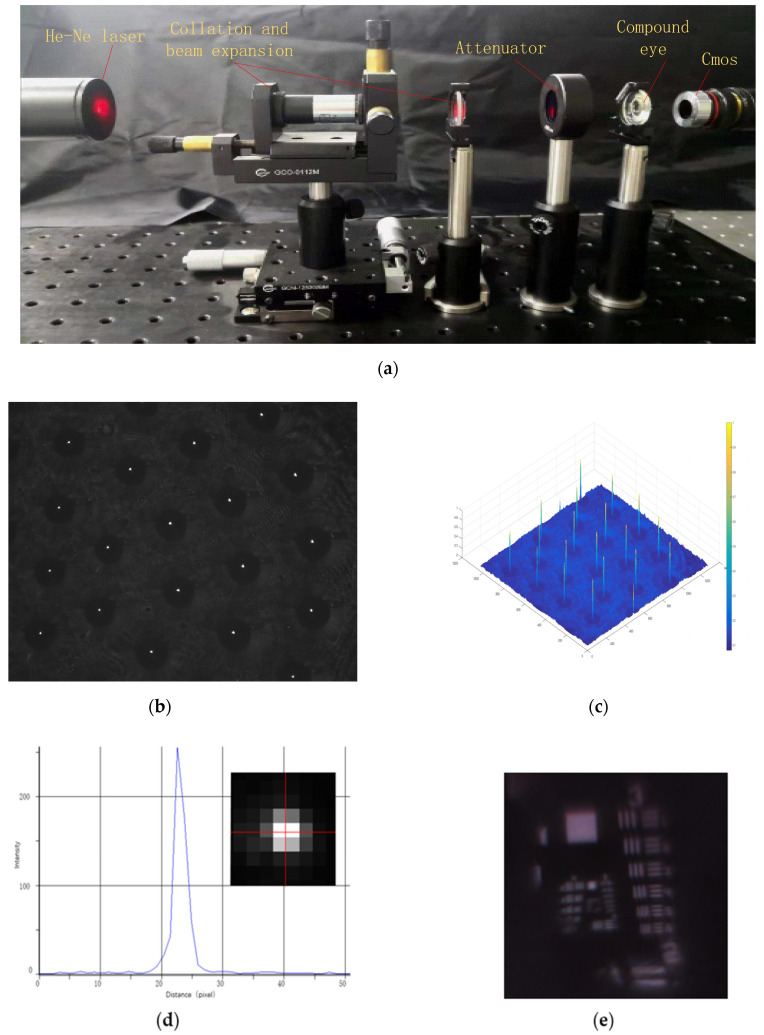
Optical characterization of the curved compound eye. (**a**) The setup to capture the focal spot; (**b**) the focal spot of each ring; (**c**) normalized intensity distribution of the curved compound eye; (**d**) the point spread function (PSF) of the central ommatidium; (**e**) the resolution target captured by the lens; (**f**) the setup to measure the field of view; (**g**) the letters captured by the camera.

**Figure 11 micromachines-12-00331-f011:**
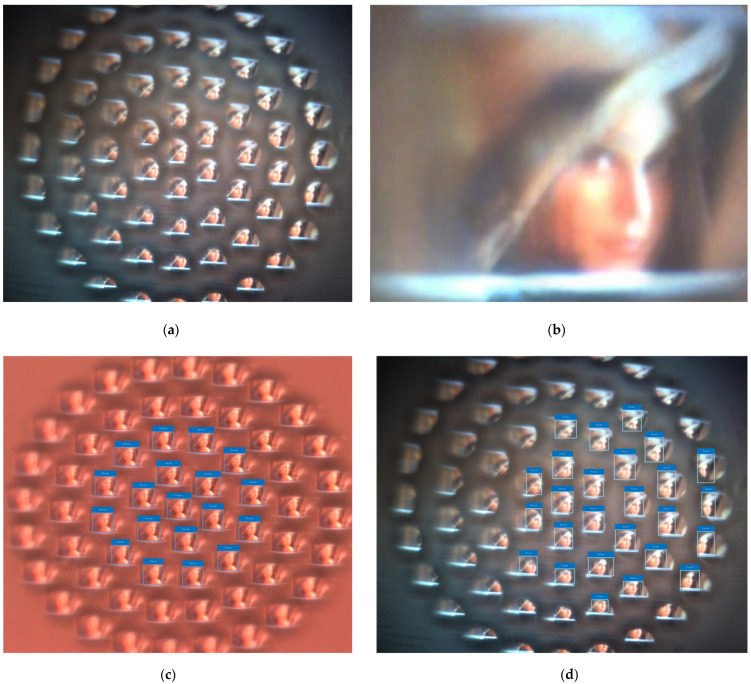
(**a**) The captured image of Lena; (**b**) the stitched image of all sub-images; (**c**) the rapid recognition result of the image of the planar MLA; (**d**) the rapid recognition result of the image of the curved compound eye; (**e**) the analysis result of the white block when it is moved 3 cm to the left; (**f**) the analysis result of the white block when it is moved 3 cm to the right.

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
