# Peer review of "Design and Integration of the Single-Lens Curved Multi-Focusing Compound Eye Camera"

_micromachines, 2021, doi:10.3390/mi12030331_

Round 1
Reviewer 1 Report
This is an interesting article about the technology of forming a bionic compound lens. The level of results in this form is sufficient to demonstrate the efficiency of the technology. I would like to draw attention to the lack of elaboration in the presentation of the results.
- Why did the authors make Fig. 1b? If they wanted to show the structure of the camera, they had to mark the individual parts and describe them in the text of the article!
- In Figure 2, the characters are too small to read at the normal scale.
- In Figure 3a, there are symbols that are unreadable at the normal scale.In equation (3), there are undescribed quantities (shown in Figure 3a)
- In my opinion, Figure 9 provides almost no information for the reader. It would be useful in Fig.9a and in Fig. 9b make enlarged inserts of one of the images.
- The authors need instead of Fig. 10a insert the optical scheme of the experiment.
- In Fig. 10g, you also need to add enlarged fragments, in this form it is unreadable.
- Figure 11ef is absolutely unreadable even at high magnification
Due to the careless presentation of the results, the article is difficult to read.
Author Response
Thank you very much for your comments that made me realize the deficiencies in the article, and I made the following changes according to your proposal.
1.The individual parts has been marked in Figure1b and introduced briefly in the article.
2.In Figure 2, the size of the characters is enlarged.
3.In Figure 3a, the symbols are enlarged. The undescribed quantities in eq3 is described.
4.Figure 9a,b are made enlarged inserts.
5.The optical parts in the picture 10a are marked to better show the experimental process.
6.The partial picture of Figure 10g is enlarged.
7.Figure 11e,f have been re-edited to increase readability.

Reviewer 2 Report
The concept of a single-lens with curved compound eye is known and has been investigated by a number of researchers. The research on fabrication process is solid and can be published with some minor improvement on presentation.
Author Response
Thank you very much for your affirmation of my work and your suggestions. The article has been moderated and improved in English by professionals. (https://www.mdpi.com/authors/english).
